# Impact of COVID-19 pandemic on emergency department visits and infant health: a scoping review protocol

Brenden Osborne [1,2] Mélika Moorjani-Houle,[2,3] Romina Fakhraei,[1,2,4] Mark Walker,[2,5,6] Shi Wu Wen [2,4,5] Yanfang Guo [4,7]

For numbered affiliations see end of article.

**Correspondence to**
Dr Yanfang Guo;
yguo@bornontario.ca

## ABSTRACT

**Introduction** The novel SARS-CoV-2 pandemic has provided a set of unique challenges for paediatric patients requiring emergency care across the globe. Reduction in paediatric emergency department (ED) usage during the COVID-19 pandemic has been widely reported, but no studies to date have consolidated and described what ramifications these reductions may have on neonatal and infant health. This scoping review aims to characterise the impact of the COVID-19 pandemic on infant ED visits and neonatal and infant health.

**Methods and analysis** A comprehensive literature search will be conducted from March 2020 to July 2022 using the following databases: Embase (Ovid), Web of Science (Clarivate Analytics), Medline (Ovid) and CINAHL (EBSCOhost). This scoping review will use a five-step framework to guide the selection, extraction and analysis of data from eligible studies, with an additional sixth step for clinical consultation. Studies in English reporting the effect of the COVID-19 pandemic on infant ED visits, as well as neonatal and infant health, will be included for screening. Key findings will be reported according to the Preferred Reporting Items for Systematic Reviews and Meta-Analyses Extension for Scoping Reviews.

**Ethics and dissemination** Research ethics board approval will not be required due to the nature of the study design. The results of this scoping review will be disseminated through publication in a peer-reviewed journal and presentation at academic conferences.

## STRENGTHS AND LIMITATIONS OF THIS STUDY

⇒ Peer-reviewed framework for scoping reviews (Arksey and O'Malley) was used to design this study.
⇒ Selected studies will compare data prior to and during the COVID-19 pandemic to assess trend differences.
⇒ A clinical expert will be consulted to inform the applicability of study results.
⇒ Availability of relevant data in selected studies may vary per site.

Several studies report a steep decrease in paediatric ED visits since the COVID-19 pandemic began.[4–7] A Canadian study of 11 paediatric centres reported a decrease in ED visits by 58% during the pandemic, compared with estimated rates.[8] The literature highlights that reduced ED usage may be a result of increased lockdown measures, social distancing and fear of contracting SARS-CoV-2.[5] Data from Korea highlight the association between increasing government restrictions and a reduction in the number of monthly ED visits.[9]

Previous work detailing the effect of the H1N1 pandemic on infant health, morbidity and mortality may offer insight regarding potential effects of SARS-CoV-2 infection in this population. Studies report delays in presentation to the ED during the H1N1 pandemic (2.8±2.3 days) as well as an increase in paediatric intensive care unit (PICU) admissions during this time compared with the prior years (0.3% vs 0.1%; 95% CI 0.05% to 0.4%).[10 11] Children younger than 2 years of age with a confirmed H1N1 diagnosis were reported to have an increased risk of hospitalisation during this period (Risk Ratio 3.3; 95% CI 1.80 to 6.05).[10] One population-based study in England reported the highest case fatality rate in their infant (<1 year) population, with a fatality rate of 151 per 100 000 cases of H1N1.[12]

## INTRODUCTION

In March 2020, the WHO declared the novel coronavirus (COVID-19) outbreak a global pandemic, which continues to affect millions globally.[1] As individuals restrict their movements, undergo mandatory social distancing and work from home, hospitals around the globe are observing a reduced patient load in their emergency departments (EDs).[2] The Centers for Disease Control and Prevention reports a 42% decrease in ED visits in the USA based on weekly means, dropping from 2.1 million per week prepandemic to 1.2 million during the pandemic, with the steepest decrease reported in the paediatric patient population.[3]

To date, most studies have focused on collecting or retrospectively analysing primary data to quantify the reduction in paediatric ED usage during the COVID-19 pandemic. The literature is lacking consensus on the implications of these reductions on infant health outside the context of a confirmed COVID-19 diagnosis in the general infant population. Given what is known about previous pandemics precipitating adverse effects on infant health,[10–12] further knowledge synthesis is required to address the novel situation created by the COVID-19 pandemic. Our objective is to conduct a scoping review to characterise the effect and to better understand the impact of the COVID-19 pandemic on infant ED visits and secondarily neonatal and infant health.

## METHODS AND ANALYSIS

This scoping review will be conducted according to the methodological framework developed by Arksey and O'Malley and refined by Levac *et al* and the Joanna Briggs Institute (JBI).[13–15] The steps outlined by this framework are (1) identification of the research question, (2) identification of relevant studies, (3) selection of studies, (4) charting the data, and (5) collating, summarising and reporting the results. As recommended by Arksey and O'Malley,[13] we included an additional consultation step with a clinical expert to find additional sources of information and inform the clinical relevance and value of this review.

### Step 1: identifying the research question

To help identify the main concepts of the primary review question, the population–concept–context (PCC) framework is suggested by the JBI.[15] The specific PCC framework for this study is presented in table 1. The primary research question for this review is: what is the current evidence reporting the effect of the COVID-19 pandemic on infant ED visits and neonatal and infant health?

### Step 2: identifying relevant studies

Our search strategy and database choice will be developed in conjunction with and refined by a trained medical librarian. Four online databases will be searched: Embase (Ovid), Web of Science (Clarivate Analytics), Medline (Ovid) and CINAHL (EBSCOhost). Key search terms were developed to capture literature related to the effect of the COVID-19 pandemic on infant ED visits, as well as neonatal and infant health. Truncation and Boolean were used to narrow, widen and combine search parameters as necessary. The finalised search strategies are available in online supplemental appendix 1. The initial literature search will be carried out on 15 July 2022 and the study will be completed on 15 September 2022.

We agreed to the following eligibility criteria for the initial search:
► Type of publication: journal articles.
► Time frame: during the COVID-19 pandemic (March 2020 onwards).
► Study population: neonates (<28 days of age) and infants (<1 year of age) presenting to the ED for medical attention.
► Study design: analytical epidemiological observational study designs (ie, cohort studies, case–control studies or cross-sectional studies), analytical ecological studies (ie, time series studies), and systematic reviews with or without meta-analyses.

We will exclude case reports and series, editorials, commentaries, letters to the editor, abstracts, conference proceedings, book chapters, narrative reviews, preprint literature, non-English studies and any studies that do not compare outcome data collected during the pandemic with a time period prior to the pandemic.

### Step 3: study selection

Records from electronic database searches will be imported into Microsoft Excel to eliminate duplicates. Two reviewers (BO, MM-H) will independently screen titles and abstracts to determine study eligibility based on predefined inclusion and exclusion criteria. A second screen of the article in full text will be performed by two independent reviewers (BO, MM-H) to ensure that studies fully meet the inclusion criteria and report relevant data. Any discrepancies will be resolved by consensus or through a third reviewer (RF). The results of the screening process will be displayed by a Preferred Reporting Items for Systematic Reviews and Meta-Analyses (PRISMA) flow diagram.[16]

### Step 4: charting the data

Data will be extracted into standardised forms in Excel by two reviewers (BO, MM-H) and checked for accuracy and completeness by a third reviewer (RF). Whenever necessary, the authors of the original study will be contacted

| Table 1 | Population–concept–context |
|---------|---------------------------|
| Population | All peer-reviewed journal articles including neonates (<28 days) and infants (<1 year) will be included. |
| Concept | Literature reporting on the frequency/rate and main reasons for neonate/infant ED visits during and before the COVID-19 pandemic will be reviewed. Literature reporting on infant outcomes, infant mortality including neonatal death (<28 days) and infant death (<1 year of age), main reasons for infant mortality, infant and neonate hospitalisation, main reasons for hospital admission, paediatric or neonatal intensive care unit admission, high-dependency unit admission, and length of ED visit and/or hospital stay will also be reviewed. |
| Context | The context will be hospital ED. The time frame is before and during the COVID-19 pandemic (March 2020 and onwards). There will be no restrictions on geographical location. |

ED, emergency department.

for additional information and clarification of data. Reviewers will resolve discrepancies through consensus or consultation with another study author (RF, YG). The data charting forms are available in online supplemental appendix 2.

Extracted data will include the following: (1) bibliometric details: title, author(s), publication year and journal; (2) study details: study design, inclusion and exclusion criteria, sample size, sample characteristics, setting, sample size included in analysis, and study period; (3) primary outcome: changes in paediatric ED visits reported during and before the COVID-19 pandemic, including numbers, percentages, frequencies and reasons for change; and (4) secondary outcomes: changes in infant morbidity and mortality details, neonatal intensive care unit admission, PICU or high-dependency unit admission, infant hospitalisations, reasons for ED visit or hospital admission, and length of ED visit or hospital stay during and before the pandemic.

### Step 5: collating, summarising and reporting the results

The results from this review will be presented in tables, while pertinent bibliometric details and critical results will be described according to standardised scoping review methodologies. We will collate results exclusively from peer-reviewed journal articles and comment on heterogeneity of the reported results. Extracted data will focus on trends in infant ED usage during and before the COVID-19 pandemic, as well as the impact of the pandemic on infant and neonatal health parameters. Gaps in the literature will be highlighted and supported by a consultation with a clinical expert in the field. The PRISMA Extension for Scoping Reviews (PRISMA-ScR) checklist will be followed to report the results of this scoping review.[17]

### Step 6: expert consultation

A consultation with a clinical expert will be sought to strengthen the rigour of the scoping review. Sharing preliminary results with the clinical expert will allow for discussion related to data interpretation, clinical applicability and dissemination strategies. The clinical expert will provide academic insight beyond what is currently reported in the literature, assist with identifying additional sources of information and inform the clinical relevance of the scoping review.

### Patient and public involvement

Patients and the public were not involved in the development of this protocol.

### ETHICS AND DISSEMINATION

Ethics approval is not required by the institutional research ethics board as scoping review methodology does not consist of primary data collection. Results from this scoping review will be presented at scientific conferences and published in a peer-reviewed journal according to the PRISMA-ScR guidelines.

**Author affiliations**
[1]Children's Hospital of Eastern Ontario Research Institute, Ottawa, Ontario, Canada
[2]Clinical Epidemiology Program, Ottawa Hospital Research Institute, Ottawa, Ontario, Canada
[3]Faculty of Medicine, University of Ottawa, Ottawa, Ontario, Canada
[4]School of Epidemiology and Public Health, Faculty of Medicine, University of Ottawa, Ottawa, Ontario, Canada
[5]Department of Obstetrics and Gynecology, Faculty of Medicine, University of Ottawa, Ottawa, Ontario, Canada
[6]Department of Obstetrics, Gynecology and Newborn Care, The Ottawa Hospital, Ottawa, Ontario, Canada
[7]BORN Ontario, Children's Hospital of Eastern Ontario, Ottawa, Ontario, Canada

**Contributors** YG, SWW and MW conceptualised the study. YG, BO and RF generated and reviewed the inclusion and exclusion criteria. RF developed the search strategy in conjunction with YG, RF and BO. BO and MM-H designed the data extraction tool. BO and MM-H drafted the protocol for publication. All authors critically revised and approved the manuscript for publication.

**Funding** This research is supported by a Canadian Institutes of Health Research Foundation Grant (FDN 148438), a Children's Hospital of Eastern Ontario Research Institute Summer Studentship Award and a University of Ottawa Faculty of Medicine Summer Studentship bursary.

**Competing interests** None declared.

**Patient and public involvement** Patients and/or the public were not involved in the design, or conduct, or reporting, or dissemination plans of this research.

**Patient consent for publication** Not required.

**Provenance and peer review** Not commissioned; externally peer reviewed.

**ORCID iDs**
Brenden Osborne http://orcid.org/0000-0002-7847-9926
Shi Wu Wen http://orcid.org/0000-0002-7227-0283
Yanfang Guo http://orcid.org/0000-0003-4749-2033

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
