## [Reviewer comments · BMJ Open]

ARTICLE DETAILS

TITLE (PROVISIONAL)	Impact of COVID-19 pandemic on emergency department visits and infant health: a scoping review protocol
AUTHORS	Osborne, Brenden; Moorjani-Houle, Mélika; Fakhraei, Romina; Walker, Mark; Wen, Shi Wu; Guo, Yanfang

VERSION 1 – REVIEW

REVIEWER	Weiner, Debra Boston Children's Hospital, Emergency Medicine
REVIEW RETURNED	31-Mar-2022

GENERAL COMMENTS	Study is important and will be valuable but description of protocol is not publication worthy. Protocol is conceptually, and for the most part methodologically, appropriate and follows standard research practices for scoping reviews but is therefore not novel in its approach so is of limited value at this time. The protocol should consider that it likely that not all of the desired data will be available from each published study and that it may be challenging to contextualize results of each study given differences in COVID surge and mitigation at different sites at different times. I will look forward to a manuscript of the actual review.
--

REVIEWER	Barrett, Michael Children's Health Ireland at Crumlin, Emergency Medicine
REVIEW RETURNED	19-Apr-2022

GENERAL COMMENTS	Well done on writing an clear protocol. I have a few minor queries. As this is an evolving story may I clarify the rationale for not making the scoping review more current? Page 4: section ' Step 2: Identifying relevant studies' : Will the literature search be inclusive of the pre-print literature? I believe it would add to the clarity of the manuscript if the protocol addressed this question directly. Please remove the results (numbers) contained in Appendix 1 from each of the search strategies. Techically they are results. In the data charting form please consider the addition of identifying the level of hospital (tertiary/ secondary paediatric , mixed hospital setting) ' Study Setting' line 20-23 page 7. Have the authors considered matching data from studies to the stage/wave of the pandemic to better understand the data extracted in the context of the stage of the pandemic regionally?
--

	Consider the inclusion of 'High dependency unit' admission as a data collection point in Appendix 2.
--	--

VERSION 1 – AUTHOR RESPONSE

Response to Reviewer 1:

Comment #1: The protocol should consider that it likely that not all of the desired data will be available from each published study and that it may be challenging to contextualize results of each study given differences in COVID surge and mitigation at different sites at different times.

Response #1: We thank the reviewer for the insightful comments. We agree that not all relevant data may be available for each included study and that available data may vary per site. These comments have been addressed in the “strengths and limitations” section of the manuscript (page 2, line 44) which is “Availability of relevant data in selected studies may vary per site”.

Response to Reviewer 2:

Comment #1: As this is an evolving story may I clarify the rationale for not making the scoping review more current?

Response #1: We appreciate the reviewer’s comment. We would like to clarify that in the identification of relevant studies (page 4, table 1, line 31-32) we will include studies published from March 2020 (declaration of the COVID-19 pandemic) until current day. The literature search will be updated throughout the final manuscript preparation process to ensure the most relevant results.

Comment #2: Page 4: section 'Step 2: Identifying relevant studies': Will the literature search be inclusive of the pre-print literature? I believe it would add to the clarity of the manuscript if the protocol addressed this question directly.

Response #2: We thank the reviewer for their comment. We have clarified on page 5, line 4: section 'Step 2: Identifying relevant studies' that the study will not include pre-print literature.

Comment #3: Please remove the results (numbers) contained in Appendix 1 from each of the search strategies. Technically they are results.

Response #3: We appreciate the reviewer’s feedback. Numerical results have been removed from the search strategies presented in Appendix 1 (supplementary file).

Comment #4: In the data charting form please consider the addition of identifying the level of hospital (tertiary/secondary paediatric, mixed hospital setting) ' Study Setting' line 20-23 page 7.

Response #4: We thank the reviewer for the great suggestion, and agree that this is an important variable to consider. We have included in our data charting form under 'Study Setting' (page 7 line 24-25, supplementary appendix 2), that we will 'Specify the level of medical facility in each study included in the review'.

Comment #5: Have the authors considered matching data from studies to the stage/wave of the pandemic to better understand the data extracted in the context of the stage of the pandemic regionally?

Response #5: We thank the reviewer for their query. We had not considered this so far. However, it would be of interest to explore the data based on the stages of the pandemic, as different stages may have corresponded with varying regional restrictions and hence may have impacted pediatric ED usage and infant health trends. The body of the protocol has been updated under 'Step 4: Charting the data' to include 'study period' (page 5 line 32).

The data charting form (page 7 line 32-33 supplementary appendix 2) has been updated to include this variable as 'Specify the stage/wave of the pandemic during which each included study was conducted'.

Comment #6: Consider the inclusion of 'High dependency unit' admission as a data collection point in Appendix 2.

Response #6: We thank the reviewer for their insightful suggestion. Table 1 detailing the population-concept-context framework for the study has been updated to include admission to high dependency unit under the concepts section (page 4 line 25-26). 'Step 4: Charting the data' has been revised to include high-dependency unit admission under secondary outcomes (page 5 line 37). The data charting form (page 7 lines 46, 48, 49, 52 supplementary appendix 2) has been amended to include admission to 'high dependency unit' (HDU) alongside NICU and PICU admission data.

Response to Editorial Requests:

Comment #1: Please revise the 'Strengths and limitations of this study' section of your manuscript (after the abstract).

Response #1: We thank the Editorial Team for their feedback and suggestions. The 'Strengths and limitations' section of the manuscript has been revised to more accurately represent the strengths and limitations of our methodology (page 2, line 31-37).

Comment #2: You have indicated that you have not included patients or the public in your study design because it is not required for a scoping review. Please note that patients and the public can be

involved in all study designs, including systematic or scoping reviews. We ask that you revise your PPI statement to be in line with our Instructions for Authors.

Response #2: We thank the Editorial Team for their comment. We have revised the patient and public involvement statement on page 6 (line 9) to state that 'Patients and the public were not involved in the development of this protocol'.

Comment #3: Along with your revised manuscript, please include a copy of the PRISMA-ScR checklist for scoping reviews indicating the page/line numbers of your manuscript where the relevant information can be found. Please mark any items not relevant to a protocol as 'n/a'.

Response #3: We thank the Editorial Team for their comment. A PRISMA-ScR checklist for scoping reviews has been submitted with the revised manuscript.

Comment #4: Please include the planned start and end dates for the study in the methods section.

Response #4:

We thank the Editorial Team for their comment. We have included under 'Step 2: Identifying relevant studies' (page 4 line 48-50) that our initial search will be carried out on July 15th, 2022 and the study will be completed by September 15th, 2022.

VERSION 2 – REVIEW

REVIEWER	Barrett, Michael Children's Health Ireland at Crumlin, Emergency Medicine
REVIEW RETURNED	23-May-2022
GENERAL COMMENTS	My questions and contributions have been acknowledged and addressed.